# SEARCH DATA STRUCTURE LEARNING

## ABSTRACT

In our modern world, an enormous amount of data surrounds us, and we are rarely interested in more than a handful of data points at once. It is like searching for needles in a haystack, and in many cases, there is no better algorithm than a random search, which might not be viable. Previously proposed algorithms for efficient database access are made for particular applications such as finding the min/max, finding all points within a range or finding the k-nearest neighbours. Consequently, there is a lack of versatility concerning what we can search when it comes to a gigantic database. In this work, we propose Search Data Structure Learning (SDSL), a generalization of the standard Search Data Structure (SDS) in which the machine has to learn how to search in the database. To evaluate approaches in this field, we propose a novel metric called Sequential Search Work Ratio (SSWR), a natural way of measuring a search's efficiency and quality. Finally, we inaugurate the field with the Efficient Learnable Binary Access (ELBA), a family of models for Search Data Structure Learning. It requires a means to train two parametric functions and a search data structure for binary codes. For the training, we developed a novel loss function, the F-beta Loss. For the SDS, we describe the Multi-Bernoulli Search (MBS), a novel approach for probabilistic binary codes. Finally, we exhibit the F-beta Loss and the MBS synergy by experimentally showing that it is at least twice as better than using the alternative loss functions of MIHash and HashNet and twenty times better than with another SDS based on the Hamming radius.

## 1 INTRODUCTION

In many applications, the machines need to perform many searches in a gigantic database where the number of relevant documents is minuscule, e.g. ten in a billion. It is like searching for some needles in a haystack. In those cases, considering every document is extremely inefficient. For productivity, the search should not consider the whole database. Traditionally, this is accomplished by building a search data structure and seeking within it. Those data structures can take many forms. For example, there are tree-based structures such as the B-Tree (Bayer & McCreight, 1970), the k-d tree (Friedman et al., 1977), the R-Tree (Guttman, 1984) or the M-Tree (Ciaccia et al., 1997) to name a few. In addition to trees, KNNG (Paredes & Chávez, 2005) build a graph designed for the k-nearest neighbour search. Later approaches improve on KNNG, both for construction and search time and for the search quality itself. In those lines, there is Efanna (Fu & Cai, 2016), HNSW (Malkov & Yashunin, 2018) and ONNG (Iwasaki & Miyazaki, 2018). One of the most common types of search data structures is the hash table. It is so useful that it is implemented natively in programming languages such as Python (with the dictionary type). Hash table is often the main tool an application will use for efficiency. For example, from a short and noisy song sample, Shazam (Wang et al., 2003) can retrieve the whole song by using hash tables filled with well-designed fingerprints of each song.

Traditionally, the design of a search data structure was for a particular type of search. For example, hash tables can retrieve documents very quickly, even in gigantic databases. However, the query must be equal to the key. This requirement makes the hash table not always applicable. For instance, if the database is indexed by date and time and we seek all documents from a specific day, then it might not be optimal to query every second of that day with an equality search. B-Tree (Bayer & McCreight, 1970) was precisely introduced for applications where a range search is preferable (and faster insertion than dichotomic search is needed). Equality and range are far from being the only

types of search. For instance, the k-nearest neighbours is another well-studied type a search. Also, the subset search is a more exotic example that occurs when every queries and documents are sets and when a document is relevant if and only if it is a subset of the query. As a final example, the auto-complete function often uses a Trie data structure (De La Briandais, 1959) to suggest the end of each word.

It is easy not to realize how the problem of efficiently finding needles in a haystack was solved multiple times for specific applications. This is the ascertainment that make Search Data Structure Learning (SDSL) a significant subject. Machine Learning has been a very flexible paradigm, whether by solving multiple NLP (Natural Language Processing) tasks with a unique Transformer (Vaswani et al., 2017) or solving most Atari games with Reinforcement Learning (Mnih et al., 2013), the capacity of a single learning algorithm to perform on multiple tasks is outstanding. Search Data Structure Learning aims at developing generic learning algorithms meant for multiple types of search. Furthermore, what makes a document relevant need not to be described formally or even understood by a human. It might be k-nearest neighbour with a complex metric or something else altogether, the only thing we need for learning is a dataset. While we use the term "Search Data Structure Learning" for the first time, algorithms that fall into its paradigm already exist. The large video-hosting platform YouTube implements an SDSL algorithm (Covington et al., 2016) for its recommendation system (the user being the query and the videos being the documents).

Not having a formalized definition of what makes a document relevant and relying on Machine Learning has its challenges, the most important being the evaluation. Traditional search data structure, such as the hash table, the Trie, the B-Tree, are exact, meaning the documents retrieved contains all and uniquely the relevant documents. For comparing those exact search data structures, when possible, the comparison between two exact search data structures is made with the asymptotic time complexity (the big-O notation). However, when the search is not exact, it is unclear how to compare structures with different efficiency and exactitude. The precision at Hamming distance of 2 is an attempt to unify those two properties into a single measure specific to the context of binary encoding. However, as described below, it fails in many aspects. It might seem like it is up to the programmer to decide what is more important between the speed and the quality of the retrieved documents. For example, the recall-queries per second (Aumüller et al., 2017) helps to visually understand the trade-off between speed and quality. In section 3, we describe a reliable measure to evaluate the efficiency and quality simultaneously of any search data structure. This metric solidifies the Machine Learning subfield of Search Data Structure Learning.

This article presents the SDSL framework that brings two crucial generalization w.r.t. its predecessors (Li et al., 2011; Cayton & Dasgupta, 2008). First, it allows for dynamic databases, i.e. databases that might change or evolve after training. For example, it is plausible that a company wants to design and train a search engine ready for distribution to multiple clients without further training on each client's database. The current mindset is to retrain each time a new database is given; however, this is not feasible in many cases. Hopefully, this article motivates the research towards models that can generalize to never seen databases. Secondly, the previous framework does not support relative relations, i.e. when the relevance of a document w.r.t. a query depends on the other documents in the database. The most studied relative relations is probably the KNN task, which is relative since it is impossible to know if a document is in the k-nearest neighbour of a query without knowing the other documents. In contrast, radius search is an example of what we call an absolute relation because it is possible to know if a document is relevant to a query only by looking at the query-document pair. In this work, however, we did not introduce relative relations only for KNN. Many interesting relative relation tasks exist; for example, another rather exciting example of relative relation is the multiple supporting facts:

> A harder task is to answer questions where two supporting statements have to be chained to answer the question [...] where to answer the question "Where is the football?" one has to combine information from two sentences "John is in the playground" and "John picked up the football". (Weston et al., 2015).

In this work, we first introduce a general framework to formalize the SDSL task in which we present a novel metric to simultaneously evaluate the efficiency and quality of the search. Then, we inaugurate the field of SDSL with Efficient Learning Binary Access (ELBA) (Section 4) that describes a family of models that use a traditional search data structure and parametric functions (e.g. neural networks) to create a **discrete binary code(s)** for both the queries and documents. A reader familiar

with the field must appreciate the difficulty that has to be overcome when dealing with (semi)-discrete structure. To instantiate ELBA, we concocted the F-beta Loss used for the training and Multi-Bernoulli Search (MBS), a novel SDS technique designed for probabilistic binary codes. Finally, for comparisons, we will instantiate ELBA with other loss functions and another SDS, namely the MIHash's loss (Cakir et al., 2017), the HashNet's loss (Cao et al., 2017), and the Hamming Radius Search C.4. We will then experimentally show the F-beta Loss and MBS's advantage by putting in evidence their synergy.

## 2 RELATED WORK

In data structure terminology, the concepts of dynamic and static structures describe whether or not the structure can change via insertion, deletion or merge. In SDSL, if the database(s) used for training are not the same as the one(s) used for evaluation, then the structure has to search for documents only seen once at insertion. From a Machine Learning perspective, this is known as a One-Shot Learning task. For example, Matching Network (Vinyals et al., 2016) tries to match never seen elements together. However, applying their technique would require a database scan. Hence it is incompatible with a gigantic database. In the same vein, soft addressing (or attention) is a differentiable mechanism to select an element from many, thus compatible with gradient descent. Memory Network (Kumar et al., 2016), Neural-Turing Machine (Graves et al., 2014) and Transformer (Vaswani et al., 2017) all use some kind of soft addressing. It is interesting for training our models but cannot be used alone in SDSL. For the same reason as above, it would require considering the whole database.

Finding the k-nearest neighbour is trivial with unlimited resources. In this field, the research focuses mainly on the efficiency of both the search and the structure's construction. The exact algorithms are as efficient in higher dimensions than a random search due to the curse of dimensionality. Consequently, the focus has recently been on approximate k-nearest neighbour. The search data structure developed are mostly tree-based, such as the k-d tree (Friedman et al., 1977) or the K-Means tree (Nister & Stewenius, 2006), and graph-based, such as the KNNG (Paredes & Chávez, 2005), Efanna (Fu & Cai, 2016), HNSW (Malkov & Yashunin, 2018) or ONNG (Iwasaki & Miyazaki, 2018) just to name a few. A good resource for comparing those approaches is the ann-benchmark Aumüller et al. (2017). In this work, we generalize the problem to conceive algorithms able to learn what to search efficiently.

Efficient Learnable Binary Access, described below, encodes queries and documents into binary vectors. In this work, we will use neural networks as the encoders. Such encoders already exists in the literature. For example, CNNH (Xia et al., 2014), DPSH (Li et al., 2015), DHN (Zhu et al., 2016), GreedyHash (Su et al., 2018), PGDH (Yuan et al., 2018), HashGan (Cao et al., 2018), ADSH (Jiang & Li, 2018) or JMLH (Shen et al., 2019), just to name a few. Below we compare different loss functions, the F-beta Loss 4 the MIHash's loss Cakir et al. (2017) and HashNet's loss Cao et al. (2017).

Graph learning, introduced in Zhu et al. (2003) for semi-supervised learning, is a type of data structure learning that has shown experimentally to be a strong idea. Those models learn to do inference from graphs, sometime by generating them first. Some approaches work with static graphs (static structures) (Zhu et al., 2003; Perozzi et al., 2014; Scarselli et al., 2008; Bruna et al., 2013) while other work with dynamic graphs (dynamic structures) (Narayan & Roe, 2018; Manessi et al., 2020). While this literature does not focus on retrieval, they learn to compute using a data structure.

To put SDSL in contrast with the Learning to Search framework (Li et al., 2011). As mentioned in the introduction, it does not support dynamic databases and relative relation. It is possible to update the framework to deal with dynamic databases by taking an expectation over the databases in the retrieval quality $Q(T)$ and computational cost $C(T)$. However, it is not clear how to deal with relative relations because the selection function $T(q, x)$ is a "matching function" that does not exist for relative tasks. Generalizing the selection function by allowing it to consider the whole database (i.e. with $T(q, X)$) does not work because $T(q, X)$ could use the ranking function $s(x, q)$ on every document and nothing would penalize such exhaustive strategies since the computational cost is the number of candidates. Nevertheless, this is not the main issue. As with the framework proposed in Cayton & Dasgupta (2008), the computational cost does not consider the retrieval cost but only the size of the candidates set (divided by the number of documents in the database for the

latter framework). Those frameworks fail to quantify the work needed to retrieve the candidates. For example, while proposing the Learning to Search framework, the authors relied on timing to evaluate their model. The SDSL framework, proposed below, provides a unique quantity that quantifies both the cost of retrieval and the candidates' quality simultaneously.

Finally, while not introduced as such, an SDSL algorithm is used in NLP. In this field, many articles attempt to accelerate the training and inference of the neural network based models, in which the main bottleneck is the normalization over a large vocabulary. Morin & Bengio (2005) use a pre-computed tree and train their model to travel from the root to a leaf, where each leaf corresponds to a word. Doing so accelerates both training and inference. Latter, Mnih & Hinton (2009) proposed a way to learn the structure of the tree.

## 3 FORMALISATION OF THE PROBLEM

Let $\mathcal{Q}$ be the query universe, let $\mathcal{U}$ be the document universe, and let $\mathbb{D}$ be the set of databases, i.e. the set of all finite sets of documents. The task is formulated with a set of relations corresponding to each database $\mathcal{R} = \{\mathcal{R}_D : \mathcal{Q} \to 2^D \mid D \in \mathbb{D}\}$. This is to allow the general case where the relation is relative.

**Definition 3.1.** The relation set $\mathcal{R}$ is **absolute** if there is a match function $\mathcal{M} : \mathcal{Q} \times \mathcal{U} \to \{\text{True}, \text{False}\}$ s.t. $\forall q \in \mathcal{Q}, D \in \mathbb{D}, d \in D, \mathcal{M}(q, d) \Leftrightarrow d \in \mathcal{R}_D(q)$ otherwise, we say it is **relative**.

This definition can easily be generalized to the cases where each $\mathcal{R}_D$ is probabilistic map by using probability instead of a truth value.

For the F-beta loss to be defined in Section 4, we restrict ourselves to absolute relation sets and thus, only the query-document pair determines if the document is relevant. However, the rest of this formalization is for both relative and absolute relation sets.

The mAP is a widely used metric in information retrieval. However, it does not consider the work done to perform the ranking. An SDS could compare the query to every document in the database and have a good mAP. In SDSL, we want to monitor the quality as well as the efficiency of retrieval. The Recall Query per second (RQPS) (Aumüller et al., 2017) is also used in the ANN literature. However, it is not suitable for theoretical analysis since the results depend on the implementation and the hardware. It is possible to generalize the RQPS by changing what quantifies the amount of work done for retrieval (to something else than the number of seconds). Nevertheless, the RQPS has a parameter (k) that limits the number of candidates to generate. This parameter prevents a model from generating all documents in the database as its candidates, which would give 100% recall without any computation and, consequently, having an excellent score doing nothing. Ultimately, the parameter k is a fix to the flaw that the RQPS does not consider the precision. In SDSL, we want to legitimately compare models that might not generate the same number of candidates. To the best of our knowledge, the precision at a Hamming distance of two (p@2) is the only proposal in the literature to consolidate the search's efficiency and quality without relying on the hardware and the implementation. Obviously, this metric has many limitations. First, it is only relevant when the model transforms the queries and documents into binary codes. More importantly, it does not consider the recall quality. For example, if a query should return several relevant documents but only one relevant document (and no irrelevant ones) are within a Hamming distance of two according to the model, the p@2 score would be maximal for this query even though the system has a very poor recall. Another significant limitation is the fact that it does not weight the score w.r.t. the distance. In many contexts, the amount of work increases a thousandfold when comparing a perfect match (Hamming distance 0) and distance 2.

In this work, we present a generic metric for any SDSL task. We grounded the metric on a very pragmatic standpoint by asking what kind of strategy a programmer would use to find relevant documents in a database quickly. At first, one might consider a random search with a good matching function (e.g. a neural network). However, if the database is enormous, this strategy will give poor performances. One could then consider filtering a significant portion of the database using a search data structure, but the retrieved documents might contain multiple false positives, decreasing the precision. We believe we can have the best of both solutions by combining them, first filtering a large portion of the database with a search data structure and then using a good matching function to filter

the retrieved documents. Finally, to evaluate if the search data structure is useful, the programmer could consider the cost of searching in the structure plus the cost of a random search in the retrieved documents versus the cost of a random search in the whole database. This is the central idea of the Search Work Ratio, the precursor of the Sequential Search Work Ratio, both defined below.

**Definition 3.2.** A **relevance oracle** is an oracle capable of computing if a document is relevant.

**Definition 3.3.** The **relevance oracle cost**, noted $\mathcal{C}(N, K, k)$, is the expected number of oracle's calls needed to find $k$ relevant documents within a set containing $N$ documents where $K$ are relevant, when doing a random search without replacement.

**Lemma 3.1.** $\mathcal{C}(N, \ K, \ k) = k(N + 1)/(K + 1)$. *(Proof in appendix A.)*

As mentioned previously, we do not intend for the search data structure to produce the final results. Instead, another entity (e.g. a program or a human) should refine the retrieved documents. In many applications, mainly when the relevance function is absolute, it is conceivable that this entity is nearly perfect or, at least, significantly more precise than the SDS. Consequently, we define the cost of finding $k$ relevant documents in a set of size $N$ containing $K$ relevant documents as the relevance oracle cost $\mathcal{C}(N, \ K, \ k)$.

As a generalization, we can weigh differently when the oracle receives a relevant document versus when it does not. The weighting would consider a real refinement entity's potential errors and put different values to the false positives and false negatives. For simplicity, in this work, we will not weight the calls to the oracle.

The Search Work Ratio (SWR) is the ratio between the work done using the SDS versus the work done without using the SDS. Consequently, an SWR score of less than 1 implies that it is less costly to use the SDS and vice-versa. Furthermore, the SWR has a simple interpretation. For example, an SWR of 1/2 implies that using the SDS reduces the cost by a factor of two.

**Definition 3.4.** Let $D \in \mathbb{D}$, $R \in 2^D$, $k \in \mathbb{N}$, $\omega_0 \in \mathbb{R}$ and $\delta_0 \in 2^D$, then the **Search Work Ratio** is

$$\text{SWR}(D, \ R, \ k, \ \omega_0, \ \delta_0) = \frac{\mathcal{C}(|\delta_0|, \ |\delta_0 \cap R|, \ k) + \omega_0}{\mathcal{C}(|D|, \ |R|, \ k)} \in \mathbb{R}^+,$$

where $D$ is a database, $R$ is the relevant documents in this database, $k$ is the minimum number of documents we want to find, $\omega_0 \in \mathbb{R}^+$ is the cost of searching with the SDS, and $\delta_0$ is the candidates retrieved by the SDS. The cost could be any complexity measure, e.g. time or space. In this work, since we will work only with hash tables, $\omega_0$ will be the number of hashes computed. We assume that using the oracle has the same cost as computing a hash. The SWR has a significant flaw; it requires that the SDS find enough relevant documents. Otherwise, it is not defined. We will now slightly generalize this definition using a relevance generator to avoid this issue.

It is not rare that an SDS can be slightly modified to produce a sequence of **sets** of candidates.. For example, an approximate tree or graph search often employs a limit of nodes in the exploration. It is possible to modify those algorithms to generate candidates with an increasing number of nodes to explore.

**Definition 3.5.** Let $D \in \mathbb{D}$, $R \in 2^D$, $k \in \mathbb{N}$, $\omega \in \mathbb{R}^{\mathbb{N}}$ and $\delta \in (2^D)^{\mathbb{N}}$ s.t. $T = \min\{t \text{ s.t. } t \in \mathbb{N} \text{ and } |\cup_{i=0}^{t} \delta_i \cap R| \geq k\}$ exists, then the **Sequential Search Work Ratio** is

$$\text{SSWR}(D, \ R, \ k, \ \omega, \ \delta) = \frac{\mathcal{C}(|\delta_T|, \ |\delta_T \cap R|, \ k - |H \cap R|) + |H| + \sum_{t=0}^{T} \omega_t}{\mathcal{C}(|D|, \ |R|, \ k)} \in \mathbb{R}^+,$$

with $H = \cup_{i=0}^{T-1} \delta_i$ if $T > 0$ and $H = \emptyset$ otherwise.

The SSWR's numerator corresponds to a random search with the relevance oracle on the last generated candidates set plus the cost of considering all previously generated candidates plus the amount of work for computing all candidates sets up to $T$. The SSWR uses the relevance oracle cost only over the last sets of candidates because the generator did not found enough relevant documents before generating the last sets of candidates. Consequently, an exhaustive search with the oracle in the previous sets of candidates was performed before asking the generator to yield more candidates. The SSWR account for this exhaustive search with the $|H|$ term. Finally, the sets of candidates are intended to be mutually exclusive because this will reduce the relevance oracle cost computed over the last sets of candidates and give a better SSWR. However, it is not mandatory.

**Definition 3.6.** The **Search Data Structure Learning** (SDSL) framework consists of minimizing the expected SSWR over generators (of sets) of candidates w.r.t. database-query pairs. Formally, given a work function $\mathcal{W} : \mathcal{G}_D, \; q \mapsto (\omega_0, \; \omega_1, \; \dots) \in \mathbb{R}^{\mathbb{N}}$ where $q \in \mathcal{Q}$ and $\mathcal{G}_D$ is a generators w.r.t. a database $D \in \mathbb{D}$ the goal of SDSL is to minimize,

$$\min_{\mathcal{G}} \; \mathbb{E}_{D,q} \left[ \text{SSWR}(D, \mathcal{R}_D(q), \; k, \; \mathcal{W}(\mathcal{G}_D, q), \; \mathcal{G}_D(q)) \right].$$

We minimize the expectation over all databases to ensure the generator's quality even if the database changes, i.e. for dynamic databases. By letting the distribution over $D$ to be deterministic, we fall into the framework with a static database.

## 4 EFFICIENT LEARNABLE BINARY ACCESS

This section describes a family of models to tackle SDSL tasks: the Efficient Learnable Binary Access (ELBA). It consists of two parametric families of function $\mathcal{F}_{\mathcal{Q}}$ and $\mathcal{F}_{\mathcal{U}}$ (e.g. neural networks) called the queries and documents encoders, and a Multi-Bernoulli Search (MBS) data structure $\mathcal{S}$ that will be made explicit later. Any function from $\mathcal{F}_{\mathcal{Q}}$ and $\mathcal{F}_{\mathcal{U}}$ must have their domain on $\mathcal{Q}$ and $\mathcal{U}$, respectively, and their image in $[0, 1]^n$ to be interpreted as the parameters of a Multi-Bernoulli[1] (in its canonical form). Precisely, ELBA is specified by the following triplet $\text{ELBA} = (\mathcal{F}_{\mathcal{Q}}, \; \mathcal{F}_{\mathcal{U}}, \; \mathcal{S})$ with $\mathcal{F}_{\mathcal{Q}} = \{f_\theta : \mathcal{Q} \to [0, 1]^n \mid \theta \in \Theta_{\mathcal{Q}}\}$ and $\mathcal{F}_{\mathcal{U}} = \{f_\theta : \mathcal{U} \to [0, 1]^n \mid \theta \in \Theta_{\mathcal{U}}\}$.

Note that in the particular case where $\mathcal{Q} = \mathcal{U}$, it is possible to use the same function for the queries and the documents ($\text{ELBA} = (\mathcal{F}, \; \mathcal{S})$). We call this the **shared** variant of ELBA. As opposed to the **unshared** variant where the parametric families might be the same, but the parameters are free to be different (e.g. the same neural network but with different parameters).

Multi-Bernoulli Search (MBS) data structure is a key-value based data structure that implements insert and search. This data structure uses $M$ back-end key-value based data structures compatible with binary vectors keys. The back-end data structures $S_1, \; S_2, \; \dots, \; S_M$, must also implement $\text{insert}(S, \text{key}, \text{value})$ where $S \in \{S_1, \; S_2, \; \dots, \; S_M\}$ is the data structure into which we insert the value w.r.t the key. Similarly, the back-end structures must implement $\text{search}(S, \text{key})$, which must return the appropriate set of values. While the key given to the back-end data structures are binary vectors, the key given the MBS must be the parameters of a Multi-Bernoulli distribution of dimension $n$, i.e.

$$\text{key} = \pi = (\pi_1, \; \pi_2, \; \dots, \; \pi_n) \in [0, 1]^n.$$

For insertion, the MBS computes the $M$ most probable outcomes of the Multi-Bernoulli (which might not be unique) and uses them as the keys for inserting in a back-end data structures. For searching, the MBS computes the $T$ most probable outcomes of the Multi-Bernoulli (which, again, might not be unique) and uses them as the keys for searching in each back-end data structures. Consequently, the search performs $TM$ back-end searches. The pseudo-code is in the appendix B.2. Note that the insert method does not require $T$. Consequently, we can make the insertions and then choose T. This fact makes possible the modification of the search to generate candidates every time it searches in a back-end structure. Finally, to conform with the SDSL's framework, it is possible to generate candidates by yielding candidates each time we search in a back-end structure.

Computing efficiently the top-k most probable outcomes of a Multi-Bernoulli is not trivial. In the appendix B, we describe how to do it. Throughout this work, we will use the **Hashing Multi-Bernoulli Search** (HMBS), an implementation of the MBS that uses hash tables as its back-end data structures. An example of how inserting, searching and generating can be found in the appendix B.3.

To implement ELBA, we need a means to select a function from each parametric family. As the parametric families, we consider neural networks, which we aim to train with gradient descent. Consequently, we need a loss function. Thus in this section, we will describe the F-beta Loss, a novel loss function design to perform well with MBS.

---

[1]The Multi-Bernoulli is a random vector composed of $n$ independent but not identical Bernoulli.

We will restrain ourselves to absolute relevance function. For this reason, we will use a dataset of the form $\{(q_i,\ d_i,\ r_i)\}_{i=1}^N = (q_i,\ d_i,\ \mathcal{M}(q_i,\ d_i))\}_{i=1}^N$. The model will try to predict the Matching function $\mathcal{M}$. We will denote the matching prediction function $\widehat{\mathcal{M}}_\theta$.

Since the Multi-Bernoulli Search requires the (canonical) parameters of the Multi-Bernoulli representation of the query $\{\pi_i^q\}_{i=1}^n$ for search and of the document $\{\pi_i^d\}_{i=1}^n$ for insert (with $n$ the chosen number of Bernoulli in the Multi-Bernoulli random variables), it is primordial that the model provides such quantity in its computational pipeline. Let $f_\theta^{\mathcal{Q}}(q) = \pi^q$ and $f_\theta^{\mathcal{U}}(d) = \pi^d$ be the parametric functions for the queries and the documents, both implemented with a neural network ending with a sigmoid. Note that, depending on the case, the two neural networks might or might not share parameters. Finally, since we want to create a synergy with Hashing Multi-Bernoulli Search, we need the bits to be all equal if and only if the matching prediction function is one. Thus we define,

$$\widehat{\mathcal{M}}_\theta(q,\ d) = \prod_{i=1}^n \pi_i^q \pi_i^d + (1 - \pi_i^q)(1 - \pi_i^d),$$

which is the probability of both Multi-Bernoulli random variables are equal according to the distributions $\{\pi_i^q\}_{i=1}^n$ and $\{\pi_i^d\}_{i=1}^n$.

We define three essential quantities: the recall, the fallout, and the predicted matching marginal (pmm for short), respectively:

$$r_\theta = \mathbb{E}_{q,d|\mathcal{M}(q,d)}\left[\widehat{\mathcal{M}}_\theta(q,\ d)\right],\ \ s_\theta = \mathbb{E}_{q,d|\neg\mathcal{M}(q,d)}\left[\widehat{\mathcal{M}}_\theta(q,\ d)\right],\ \ \ \ m_\theta = \mathbb{E}_{q,d}\left[\widehat{\mathcal{M}}_\theta(q,\ d)\right].$$

We can compute empirical averages to produce unbiased estimators of those quantities.

$$\hat{r}_\theta = \frac{1}{|I^+|}\sum_{i\in I^+}\widehat{\mathcal{M}}_\theta(q_i,\ d_i),\ \ \ \ \ \hat{s}_\theta = \frac{1}{|I^-|}\sum_{i\in I^-}\widehat{\mathcal{M}}_\theta(q_i,\ d_i),\ \ \ \ \ \ \ \ \hat{m}_\theta = q\hat{r}_\theta + (1 - q)\hat{s}_\theta.$$

where $I^+$ and $I^-$ are the sets of indexes for query-document pairs in the dataset that match and do not match, respectively, and with $q$ being the probability of having a matching pair. It is also possible to derive an estimator for the precision $p_\theta = \frac{qr_\theta}{m_\theta}$, with $\hat{p}_\theta = \frac{q\hat{r}_\theta}{\hat{m}_\theta}$. However, it is biased.

For numerical stability and because the gradients w.r.t $r_\theta$, $s_\theta$ and $m_\theta$ are near zero when the training starts, we need to consider their logarithm for the loss function. Maximizing precision gives a model capable of discriminating between positive and negative pairs, but it leaves the recall untouched, and having a high recall is vital for the HMBS to find the relevant documents. Furthermore, only maximizing the recall induce the model towards a constant Multi-Bernoulli distribution with zero entropy, i.e., independent of the input and where all the probabilities are near 0 or 1. We tried maximizing the recall while minimizing the fallout, i.e. with a loss similar to $\max_\theta \log(r_\theta) - \lambda \log(s_\theta)$. However, we found it extremely hard to optimize — there was no sweet spot for lambda. When the model's recall was sufficient, it was because it collapsed to a constant function. Scheduling $\lambda$ to alternate between a small value and a relatively high value has shown limited experimental success. In the end, we were looking for a tradeoff between the precision and the recall. The F-beta came naturally. However, since our precision estimator is biased, it is simpler to reparameterize the standard F-beta with the pmm using $p_\theta = \frac{qr_\theta}{m_\theta}$ given us,

$$F_\beta = \frac{(1 + \beta^2)qr_\theta}{q\beta^2 + m_\theta}\ \ \ \text{with } \beta \in \mathbb{R}^+$$

For the above reasons, we considered the logarithm of the F-beta,

$$\log F_\theta = \log((1 + \beta^2)q) + \log(r_\theta) - \log(q\beta^2 + m_\theta)\ \ \ \text{with } \beta \in \mathbb{R}^+$$

However, if we replace the recall term by its estimator directly we will get the LogSumExp (LSE) w.r.t. $\log\widehat{\mathcal{M}}_\theta(q_i,\ d_i)$ for $i \in I^+$. i.e., $\log(\hat{r}_\theta) = \text{LSE}\{\log\widehat{\mathcal{M}}_\theta(q_i,\ d_i) \mid i \in I^+\}$ which is known to act as a soft maximum (not to be confused with its gradient, the Softmax). Doing this will yield a near-zero gradient for the matching pairs with the lowest predicted matching value. It would be problematic since those pairs are the ones that need the most gradient.

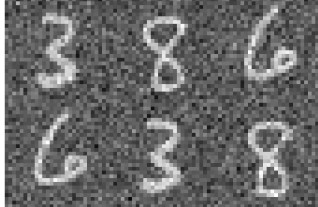 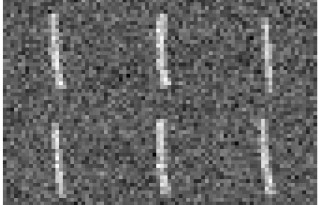

Figure 1: In the first row are the queries and in the second row are the documents. Each query corresponds uniquely to one document. On the left, the task is easy, but on the right, it is not.

Instead, we propose an alternative estimator of the log F-beta, which we call the F-beta Loss, where we replace the LSE with the average logarithm of the predicted matching value.

$$\log \widehat{F}_\theta = c + \frac{1}{|I^+|} \sum_{i \in I^+} \log \widehat{\mathcal{M}}_\theta(q_i, \, d_i) - \log(q\beta^2 + \hat{m}_\theta)$$

with $\beta \in \mathbb{R}^+$ and $c = \log(1 + \beta^2) + \log q$.

Note that it is simple to compute with numerical stability the logarithm of the sigmoid function. Most, if not all, Machine Learning libraries natively define this function.

## 5 EXPERIMENTS AND RESULTS

We performed all experiments on a dataset build from MNIST (LeCun et al., 1998), which we call **NoisyMnist**. We do not intend NoisyMnist to be a challenging task but rather a tool to analyze the convergence properties and draw comparisons between each models' qualities. In this dataset, the document and query are MNIST images with value ranging from 0 to 1 with additive Gaussian noise (the value can consequently go below 0 and above 1). The mean and std of the noise is 0 and 0.2, respectively. The relevance function of NoisyMnist is absolute, and we define the matching function as follows: a query match with a document if and only if their original MNIST image was the same before we added the noise. In figure 1, there are 6 examples of queries-documents pairs. For evaluation, We build a fixed database with 10K different images from MNIST not accessible while training. From those 10K, we randomly selected 1K to create the queries. Finally, we added the noise on each image (the query and their corresponding document do not share the same noise). In this evaluation database, there is a unique document relevant for each query. It is one in ten thousand, making it a proper database for SDSL.

We considered two alternatives for the F-beta Loss and one alternative to the MBS. For the F-beta Loss, we selected the loss function of MIHash (Cakir et al., 2017) and the loss function of HashNet (Cao et al., 2017) because of their compatibility with with ELBA. More specifically, they both produce quantities that can be interpreted as the parameters of a Multi-Bernoulli, and they both can be trivially generalized to the unshared case. Furthermore, as an alternative to MBS, we choose the Hamming Radius Search (HRS) described in appendix C.4. Combining the three losses with the two data structures creates six Efficient Learnable Binary Access models. We deployed each model in both the shared and unshared categories, for a total of twelve scenarios.

Training was run 5 times for each model, and the top 5 sets of parameters w.r.t the SSWR were selected for a total of 25 sets of parameters for each of the twelve models. All the values provided are the average of those 25 points. Each training consisted of 100K batches of size 32, which was plenty for all models to converge. At each 500 batches, we performed an evaluation giving us 200 sets of parameters to select from. We performed all evaluations using the same fixed set of 10K validation documents and a corresponding fixed set of 1K validation query described above. Those documents and queries were never seen while training. All networks are ResNet18 (He et al., 2016) adapted to MNIST, i.e. the first convolution takes from one channel with no stride and the last linear layer output a vector of size 64 (for 64 bits). The hyperparameters and training schedule can be found in the appendix C. Finally, we use halting for all six models as is describe below.

Table 1: The comparative table of SSWR

| | Unshared | | Shared | |
|---|---|---|---|---|
| Models | HRS | HMBS | HRS | HMBS |
| F-beta | 0.3798 | **0.0169** | 0.1636 | **0.0035** |
| MIHash | 1.3559 | 1.8907 | 0.2119 | 0.0083 |
| HashNet | 1.4162 | 2.0001 | 0.2536 | 0.2828 |

When using generators iterating over binary codes, such as HRS and MBS, it is crucial to halt the iteration before it finishes. For example, for 64 bits, there are $2^{64}$ possible hashes, which is certainly larger than any database. In those cases, it does not make sense to compute every possible hash. **Halting** is the mechanism that decides when to stop a relevance generator and produce the database's remaining document as the final candidates set. Note that, when halting, the SSWR is always greater than one. In all experiments with HRS, we used a halting of 2081. It corresponds to generating every document with binary codes within a radius of two from the query's code (for 64 bits). In this case, halting afterward is arbitrary since the next 41664 codes (distance of 3) come in no particular order. Finally, in all experiments with HMBS, we use a halting of 5001, which corresponds to stopping when the amount of work done exceeds the expected amount of work without an SDS, i.e. $\mathcal{C}(10000, \ 1, \ 1)$.

Table 1 shows that F-beta (from this work) outperforms MIHash and HashNet in both unshared and shared categories. Noteworthily, MIHash and HashNet fail in the unshared categories. While Hash-Net successfully produces binary codes with lower Hamming distance for positive than negative pairs, the distance is way too high to be used with hash tables. MIHash, on the other hand, only push towards increasing the mutual information between the Hamming distance and whether or not the pair matches. It implies that there is no (explicit) pressure towards having a small Hamming distance. The synergy between the loss function and shared parameters is why MIHash and Hash-Net produce low Hamming distances for positive pairs for the shared problem. The found solutions for the unshared problem are not available when the queries and documents networks are the same. Parameter sharing acts as a colossal regularization. Those poor results suggest that both MIHash and HashNet are constrained to similarity search. Making F-beta the superior loss function for ELBA with hash table based SDS.

On the other hand, HMBS's results are far superior to those of HRS. In the case of shared F-beta, they are 46 times better. The only case for which it is not true is for shared HashNet. However, both of those models are not competitive. The fat tail of its Hamming distances distribution is at cause. In our experiments, we noted that varying the halting did not seem to change the results drastically. Additional figures provided in the appendix D might convey a more intuitive understanding of the contrast between the different scenarios.

## 6 CONCLUSION AND FUTURE WORK

In this article, we proposed a novel and practical field of Machine Learning called Search Data Structure Learning, for which we propose a natural metric, the SSWR. We inaugurated this field with a new family of models, the Efficient Learnable Binary Access, which we instantiated with the F-beta Loss and the MBS that outperformed multiple alternatives, reducing by at least a factor of two the SSWR. We cannot overstate the importance of F-beta in this project. The capacity to obtain convergence on discrete output without being caught in a local minimum is a powerful tool. In NLP, the F-beta led us to several exciting breakthroughs when learning semantically rich discrete embedding for words. The fact that studying SDSL led to breakthroughs in other domains is, for us, an attestation of its significance.

Furthermore, we plan to extend the formalization to consider insertion, deletion, dependent queries, and dependent retrieved documents in future work. Such generalization could be useful for tasks like dialogue modelling or question answering. Also, in this work, the halting procedure was simplistic. In the future, we are interested in working with models that can decide when to stop based on the query and the retrieved documents. Finally, we are eager to work with non-hashing-based approaches, such as trees or graphs.

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

## A  THE RELEVANCE ORACLE COST

Formally, let $S$ be a set of $N$ documents containing $K \leq N$ relevant documents. From this $K$ elements, we want to find at least $k \leq K$ elements. We want to compute how many calls in expectation are needed to find those $k$ elements if we sample from $S$ uniformly without replacement. We will denote this expectation $\mathcal{C}(N, K, k)$.

Let $\mathcal{G}(k \; ; \; N, K, n)$ be the Hypergeometric Distribution with parameters $N$, $K$, $n$. This distribution gives the probability that from a set with $N$ documents from which $K$ are relevant, we sample exactly $k$ relevant documents in $n$ uniform trials without replacement.

$$\mathcal{G}(k \; ; \; N, K, n) = \frac{\binom{K}{k}\binom{N-K}{n-k}}{\binom{N}{n}}.$$

Let $\mathcal{P}(n \; ; \; N, K, k)$ be probability distribution, defined below, with parameters $N$, $K$, $k$. This gives the probability that from a set with $N$ documents from which $K$ are relevant, it takes $n$ uniform trials without replacement to get precisely $k$ relevant documents. We have,

$$\mathcal{P}(n \; ; \; N, K, k) = \mathcal{G}(k - 1 \; ; \; N, K, n - 1)\frac{K - (k-1)}{N - (n-1)},$$

which is the probability that we have $k - 1$ relevant documents in $n - 1$ trials multiplied by the probability that we sample a relevant document in a set with $N - (n-1)$ documents from which $K - (k-1)$ are relevant.

Finally, the expectation of $\mathcal{P}(n \; ; \; N, K, k)$ yields the wanted measure. I.e. if $X \sim \mathcal{P}(\cdot \; ; \; N, K, k)$ then,

$$\mathcal{C}(N, \; K, \; k) = \mathbb{E}\left[X\right] = \sum_{n=1}^{N} n\mathcal{P}(n \; ; \; N, K, k) = \left(\frac{N+1}{K+1}\right)k,$$

with the last equality shown below A.1.

**Lemma A.1.** *If $X \sim \mathcal{P}(\cdot \; ; \; N, K, k)$ then,*

$$\mathbb{E}\left[X\right] = \left(\frac{N+1}{K+1}\right)k$$

*Proof.* with $A \geq a$, $B \geq b$ and $A \geq B$ all natural numbers.

$$
\begin{aligned}
\mathbb{E}\left[X\right] &= \sum_{n=1}^{N} n \mathcal{P}(n \; ; \; N, K, k) \\
&= \sum_{n=k}^{N-(K-k)} n \mathcal{P}(n \; ; \; N, K, k) && \text{remove zeros} \\
&= \sum_{n=k}^{N-(K-k)} n \mathcal{G}(k-1 \; ; \; N, K, n-1) \frac{K-(k-1)}{N-(n-1)} \\
&= \sum_{n=k}^{N-(K-k)} n \frac{\binom{K}{k-1}\binom{N-K}{n-k}}{\binom{N}{n-1}} \frac{K-(k-1)}{N-(n-1)} \\
&= \sum_{n=k}^{N-(K-k)} n \frac{\frac{k}{K-k+1}\binom{K}{k}\binom{N-K}{n-k}}{\frac{n}{N-n+1}\binom{N}{n}} \frac{K-(k-1)}{N-(n-1)} && \text{with } \binom{A}{a-1} = \frac{a}{A-a+1}\binom{A}{a} \\
&= \sum_{n=k}^{N-(K-k)} k \frac{\binom{K}{k}\binom{N-K}{n-k}}{\binom{N}{n}} \\
&= \sum_{n=k}^{N-(K-k)} k \frac{\binom{n}{k}\binom{N-n}{K-k}}{\binom{N}{K}} && \text{with } \frac{\binom{B}{b}\binom{A-B}{a-b}}{\binom{A}{a}} = \frac{\binom{a}{b}\binom{A-a}{B-b}}{\binom{A}{B}} \\
&= \frac{k}{\binom{N}{K}} \sum_{n=k}^{N-(K-k)} \binom{n}{k}\binom{N-n}{K-k} \\
&= \frac{k}{\binom{N}{K}} \binom{N+1}{K+1} && \text{a variant of Vandermonde's identity} \\
&= \frac{k}{\binom{N}{K}} \frac{N+1}{K+1} \binom{N}{K} && \text{with } \binom{A+1}{a+1} = \frac{A+1}{a+1}\binom{A}{a} \\
&= \left(\frac{N+1}{K+1}\right) k
\end{aligned}
$$

$\square$

## B  TOP-K MAXIMAL MULTI-BERNOULLI OUTCOMES

Let $b_i$ be $n$ independent and non-identical Bernoulli random variables with probability $\pi_i$ to be one. Let $z_i$ be the most probable outcome of $b_i$ and let $p_i$ be the probability that $b_i$ be the most probable outcome ($b_i = z_i$) i.e.

$$
z_i = \begin{cases} 1 & \text{if } \pi_i > 1/2 \\ 0 & \text{otherwise} \end{cases} \qquad\qquad p_i = \begin{cases} \pi_i & \text{if } \pi_i > 1/2 \\ 1-\pi_i & \text{otherwise} \end{cases}
$$

note that $p_i$ is always greater or equal than $1/2$. The theorem B.2 gives us the following result.

$$
P(b=x) > P(b=y) \iff \sum_{\substack{i=1 \\ \text{s.t. } x_i \neq z_i}}^{n} \log(p_i) - \log(1-p_i) < \sum_{\substack{i=1 \\ \text{s.t. } y_i \neq z_i}}^{n} \log(p_i) - \log(1-p_i)
$$

This implies that finding the Top-K Minimal Subset Sum of $A = \{a_1, \; a_2, \; \ldots, \; a_n\}$ with $a_i = \log(p_i) - \log(1-p_i) \geq 0$ will yield the Top-K Maximal Multi-Bernoulli Outcomes. The Top-K Minimal Subset Sum problem can be solve using standard programming techniques. The pseudo-code for the reduction is in the appendix B.2.

### B.1 REDUCTION TO TOP-K MINIMAL SUBSET SUM

**Lemma B.1.** *Let $p_i \in [0, 1]$, $a_i \in \{0, 1\}$ and $b_i \in \{0, 1\}$, for $i \in \{0, 1, \ldots, n\}$ then*

$$\prod_{i=1}^{n} p_i^{a_i}(1 - p_i)^{1-a_i} < \prod_{i=1}^{n} p_i^{b_i}(1 - p_i)^{1-b_i}$$
$$\Longleftrightarrow$$
$$\prod_{i=1}^{n} p_i^{1-a_i}(1 - p_i)^{a_i} > \prod_{i=1}^{n} p_i^{1-b_i}(1 - p_i)^{b_i}$$

*Proof.* Let $I^+ = \{i \mid i \in \mathbb{N}, \ 1 \leq i \leq n, \ a_i = b_i\}$ and let $I^+ = \{i \mid i \in \mathbb{N}, \ 1 \leq i \leq n, \ a_i \neq b_i\}$ we have,

$$\prod_{i=1}^{n} p_i^{a_i}(1 - p_i)^{1-a_i} < \prod_{i=1}^{n} p_i^{b_i}(1 - p_i)^{1-b_i}$$

$$\Longleftrightarrow \prod_{i \in I^+} p_i^{a_i}(1 - p_i)^{1-a_i} \prod_{i \in I^-} p_i^{a_i}(1 - p_i)^{1-a_i} < \prod_{i \in I^+} p_i^{b_i}(1 - p_i)^{1-b_i} \prod_{i \in I^-} p_i^{b_i}(1 - p_i)^{1-b_i}$$

$$\Longleftrightarrow \prod_{i \in I^-} p_i^{a_i}(1 - p_i)^{1-a_i} < \prod_{i \in I^-} p_i^{b_i}(1 - p_i)^{1-b_i}$$

$$\Longleftrightarrow \prod_{i \in I^-} p_i^{1-b_i}(1 - p_i)^{b_i} < \prod_{i \in I^-} p_i^{1-a_i}(1 - p_i)^{a_i} \qquad \text{since } \forall i \in I^-, \ a_i = 1 - b_i$$

$$\Longleftrightarrow \prod_{i \in I^+} p_i^{1-b_i}(1 - p_i)^{b_i} \prod_{i \in I^-} p_i^{1-b_i}(1 - p_i)^{b_i} < \prod_{i \in I^+} p_i^{1-a_i}(1 - p_i)^{a_i} \prod_{i \in I^-} p_i^{1-a_i}(1 - p_i)^{a_i}$$

$$\Longleftrightarrow \prod_{i=1}^{n} p_i^{1-b_i}(1 - p_i)^{b_i} < \prod_{i=1}^{n} p_i^{1-a_i}(1 - p_i)^{a_i}$$

$\square$

**Theorem B.2.** *Let $b$ be a Multi-Bernoulli of parameter $\pi \in \,]0, 1[^n$ and let*

$$z_i = \begin{cases} 1 & \text{if } \pi_i > 1/2 \\ 0 & \text{otherwise} \end{cases} \qquad\qquad p_i = \begin{cases} \pi_i & \text{if } \pi_i > 1/2 \\ 1 - \pi_i & \text{otherwise} \end{cases}$$

*then, for any $x \in \{0, 1\}^n$ and $y \in \{0, 1\}^n$ we have*

$$P(b{=}x) > P(b{=}y) \Longleftrightarrow \sum_{\substack{i=1 \\ s.t. \ x_i \neq z_i}}^{n} \log(p_i) - \log(1 - p_i) < \sum_{\substack{i=1 \\ s.t. \ y_i \neq z_i}}^{n} \log(p_i) - \log(1 - p_i)$$

*Proof.* For notational simplicity, let $\bar{x}_i = x_i \oplus z_i$ and let $\bar{y}_i = y_i \oplus z_i$ (with $\oplus$ being the exclusive or)

$P(b{=}x) > P(b{=}y)$

$$\iff \quad \prod_{i=1}^{n} P(b_i{=}x_i) > \prod_{i=1}^{n} P(b_i{=}y_i) \qquad \text{since the Bernoulli are independent}$$

$$\iff \quad \prod_{i=1}^{n} \pi_i^{x_i}(1-\pi_i)^{1-x_i} > \prod_{i=1}^{n} \pi_i^{y_i}(1-\pi_i)^{1-y_i}$$

$$\iff \quad \prod_{i=1}^{n} p_i^{1-\bar{x}_i}(1-p_i)^{\bar{x}_i} > \prod_{i=1}^{n} p_i^{1-\bar{y}_i}(1-p_i)^{\bar{y}_i} \quad \text{proved by cases } (\pi_i > 1/2 \text{ and } \pi_i \leq 1/2)$$

$$\iff \quad \prod_{i=1}^{n} p_i^{\bar{x}_i}(1-p_i)^{1-\bar{x}_i} < \prod_{i=1}^{n} p_i^{\bar{y}_i}(1-p_i)^{1-\bar{y}_i} \qquad \text{by lemma B.1}$$

$$\iff \quad \sum_{i=1}^{n} \bar{x}_i \log(p_i) + (1-\bar{x}_i)\log(1-p_i) < \sum_{i=1}^{n} \bar{y}_i \log(p_i) + (1-\bar{y}_i)\log(1-p_i)$$

$$\iff \quad \sum_{i=1}^{n} \bar{x}_i \left(\log(p_i) - \log(1-p_i)\right) + \log(1-p_i) <$$

$$\sum_{i=1}^{n} \bar{y}_i \left(\log(p_i) - \log(1-p_i)\right) + \log(1-p_i)$$

$$\iff \quad \sum_{i=1}^{n} \bar{x}_i \left(\log(p_i) - \log(1-p_i)\right) < \sum_{i=1}^{n} \bar{y}_i \left(\log(p_i) - \log(1-p_i)\right)$$

$$\iff \quad \sum_{\substack{i=1 \\ \text{s.t. } \bar{x}_i=1}}^{n} \log(p_i) - \log(1-p_i) < \sum_{\substack{i=1 \\ \text{s.t. } \bar{y}_i=1}}^{n} \log(p_i) - \log(1-p_i)$$

$\square$

*Remark.* The case where $\pi_i$ is exactly one or zero for some $i$ can be trivially taken into account by setting the bit to the only possible value. All outcomes that are not generated will have probability zero.

## B.2 ALGORITHMS

---
**Algorithm 1** Multi-Bernoulli Search Data Structure - insert
---
**Require:** Int $n$, Int $M$, SDSArray[M] $S$, RealArray[n] key, Object value
  outcomes $\leftarrow$ Top-K Maximal Multi-Bernoulli Outcomes($M$, $n$, key)
  **for** $i \in \{1, \ldots, M\}$ **do**
    insert($S_i$, outcomes$_i$, value)
  **end for**
---

---
**Algorithm 2** Multi-Bernoulli Search Data Structure- search
---
**Require:** Int $n$, Int $M$, Int $T$, SDSArray[M] $S$, RealArray[n] key
  values $\leftarrow$ set()
  outcomes $\leftarrow$ Top-K Maximal Multi-Bernoulli Outcomes($T$, $n$, key)
  **for** $j \in \{1, \ldots, T\}$ **do**
    **for** $i \in \{1, \ldots, M\}$ **do**
      values $\leftarrow$ values $\cup$ search($S_i$, outcomes$_j$)
    **end for**
  **end for**
  **return** values
---

---

**Algorithm 3** Top-K Maximal Multi-Bernoulli Outcomes

---

**Require:** Int $k$, Int $n$, RealArray$[n]$ $\pi$
1: $z \leftarrow$ BinaryArray$[n]$
2: $p \leftarrow$ RealArray$[n]$
3: $a \leftarrow$ RealArray$[n]$
4: **for** $i = 1, \ldots, n$ **do**
5:    **if** $\pi_i > 1/2$ **then**
6:       $z_i \leftarrow 1$
7:       $p_i \leftarrow \pi_i$
8:    **else**
9:       $z_i \leftarrow 0$
10:      $p_i \leftarrow 1 - \pi_i$
11:    **end if**
12:    $a_i \leftarrow \log(p_i) - \log(1 - p_i)$
13: **end for**
14: outcomes $\leftarrow$ BinaryArray$[k \times n]$
15: index $\leftarrow$ TopKMinimalSubsetSumIndexes($k$, $n$, $a$)
16: **for** $j = 1, \ldots, k$ **do**
17:    **for** $i = 1, \ldots, n$ **do**
18:       **if** $i \in$ indexes$_j$ **then**
19:          outcomes$_{ji} \leftarrow 1 - z_i$
20:       **else**
21:          outcomes$_{ji} \leftarrow z_i$
22:       **end if**
23:    **end for**
24: **end for**
25: **return** outcomes

---

### B.3 HMBS EXAMPLE

In this example, the number of bits in the Multi-Bernoulli is $n = 3$, the number of back-end data structures is $M = 3$, and the number of search keys is $T = 2$. Since this is a Hashing Multi-Bernoulli Search data structure, the $M = 3$ back-end structures will be hash tables; we will call them $H_1$, $H_2$ and $H_3$.

Let say we want to insert a document with the following key,

$$\pi_1 = (0.3, 0.1, 0.8).$$

We first need to compute its $M = 3$ most probable outcomes. Here they are in order:

$$(0, 0, 1), \ (1, 0, 1), \ (0, 0, 0)$$

then we will insert the document in $H_1$ using the most probable outcome as the key, insert the document in $H_2$ with the second most probable outcome as the key and insert the document in $H_3$ with the third most probable outcome as the key.

Now, let say we have two other documents to insert with the following keys, respectively:

$$\pi_2 = (0.7, 0.9, 0.2), \ \pi_3 = (0.3, 0.2, 0.1).$$

Here are their $M = 3$ most probable outcomes, respectively:

$$(1, 1, 0), \ (0, 1, 0), \ (1, 1, 1) \text{ and } (0, 0, 0), \ (1, 0, 0), \ (0, 1, 0).$$

We will then insert both of them in the $M = 3$ hash tables like we did for the first document.

To search in the HMBS given a query $\pi = (0.1, 0.6, 0.2)$, we first need to compute its $T = 2$ most probable outcomes:

$$(0, 1, 0), \ (0, 0, 0).$$

We will then search in the $M = 3$ hash tables with these $T = 2$, doing a total of $TM = 6$ search in the back-end structures. These searches will find all of the three documents since the first document

---

is in $H_3$ with the key $(0, 0, 0)$, the second document is in $H_2$ with the key $(0, 1, 0)$ and the third document is both in $H_1$ with the key $(0, 0, 0)$ and in $H_3$ with the key $(0, 1, 0)$.

To generate documents as in the SDSL's framework, we do not need the parameter $T$. However, we might want to halt before considering all possible outcomes of the query. With the same query as above, say we want to generate document but halt after the fifth hashes, i.e. after doing 5 searches in the back-end structures. We will first generate the most probable outcome $(0, 1, 0)$ and search in $H_1$; however, we will find nothing. We will then try the same outcome in $H_2$ and find the second document, which we will yield. Then we will try the same outcome in $H_3$ and find the third document, which we will yield. After this, we will compute the second most probable outcome $(0, 0, 0)$ and search in $H_1$ to find the third document again; thus, we will do nothing. After we will try the same outcome in $H_2$ and find nothing. Finally, we will halt since we computed 5 hashes, ultimately never finding the first documents. Note that if there were multiple documents simultaneously, we would have yielded them together as a set.

## C    EXPERIMENTS' MODELS

### C.1    FBETA

For the F-beta model 4 we use ramping for the $\beta$ hyperparameter using this equation,

$$\log_2 \beta_i = \begin{cases} i\frac{32-8}{10K} - 32 & \text{if } i < 10K \\ -8 & \text{otherwise} \end{cases}$$

for each batch $i = 0, \ldots, 100K$.

### C.2    MIHASH

MIHash Cakir et al. (2017) is based on the mutual information

$$\mathcal{I}(X, Y) = \sum_{z \in \Omega} P(X{=}z,\ Y{=}z) \log \left( \frac{P(X{=}z,\ Y{=}z)}{P(X{=}z)P(Y{=}z)} \right)$$

and a generalization of the Hamming distance,

$$d(x, y) = \frac{1}{2}(n - x \cdot y)$$

for $x$ and $y$ in $\mathbb{R}^n$. Note that if $x$ and $y$ are in $\{0, 1\}^n$, $d(2x - 1, 2y - 1)$ is their Hamming distance.

Lets use the above notation, i.e. let $f_\theta^{\mathcal{Q}}(q) = \pi^q \in [0, 1]^n$ and $f_\theta^{\mathcal{U}}(d) = \pi^d \in [0, 1]^n$ be the parametric query and document functions (in the original article, they uses the same function for the queries and the documents). Let $X$ and $Y$ be two Multi-Bernoulli of dimension $n$ with parameters $\pi^q$ and $\pi^d$ respectively. Finally, with $H = d(2X - 1,\ 2Y - 1)$, they aim to maximize,

$$\mathcal{I}(H, \mathcal{M}(q, d))$$

To allow gradient descent, they use differentiable histogram binning, i.e. they approximate $P(H{=}k \mid \mathcal{M}(q, d){=}1)$ with

$$P(H{=}k \mid \mathcal{M}(q, d){=}1) = \frac{1}{|I^+|} \sum_{i \in I^+} \delta_{i,k}$$

with

$$\delta_{i,k} = \begin{cases} d_i - (k - 1) & \text{if } d_i \in [k - 1, k] \\ (k + 1) - d_i & \text{if } d_i \in [k, k + 1] \\ 0 & \text{otherwise} \end{cases}$$

and $d_i = d(2\pi^{q_i} - 1,\ 2\pi^{d_i} - 1)$. Similarly they estimate $P(H{=}k \mid \mathcal{M}(q, d){=}0)$ and $P(H{=}k)$ to compute the mutual information.

### C.3 HASHNET

HashNet Cao et al. (2017) optimize an increasingly closer to discrete sequence of tasks to alleviate the challenge of solving a discrete task with differentiable methods. It is possible to frame their approach entirely with sigmoids but it is is simpler to use the tanh function as in the original article. Let $\text{Net}_\theta^\mathcal{Q}(q) = \text{logits}^q \in \mathbb{R}^n$ and $\text{Net}_\theta^\mathcal{U}(d) = \text{logits}^d \in \mathbb{R}^n$ be the parametric query and document functions before activation. Let $g_\theta^\mathcal{Q}(q) = \tanh\left(\beta \text{Net}_\theta^\mathcal{Q}(q)\right)$ and $g_\theta^\mathcal{U}(d) = \tanh\left(\beta \text{Net}_\theta^\mathcal{U}(d)\right)$ be the activated functions. Note that,

$$f_\theta^\mathcal{Q}(q) = \frac{g_\theta^\mathcal{Q}(q) + 1}{2} \qquad\qquad f_\theta^\mathcal{U}(d) = \frac{g_\theta^\mathcal{U}(d) + 1}{2}$$

For simplicity, let $g_i^\mathcal{Q} = g_\theta^\mathcal{Q}(q_i)$ and $g_i^\mathcal{U} = g_\theta^\mathcal{U}(d_i)$. In their work they modelize the matching random variable with,

$$\mathcal{M}(q_i,\ d_i) \sim \text{Ber}\left(\sigma\left(\alpha\ g_i^\mathcal{Q} \cdot g_i^\mathcal{U}\right)\right)$$

This gives,

$$P(\mathcal{M}(q_i,\ d_i){=}m) = \sigma\left(\alpha\ g_i^\mathcal{Q} \cdot g_i^\mathcal{U}\right)^m \left(1 - \sigma\left(\alpha\ g_i^\mathcal{Q} \cdot g_i^\mathcal{U}\right)\right)^{1-m}$$

Finally, they train the model with the (weighted) negative log-likelihood.

$$J_i = w_i \left(\log\left(1 + \exp\left(\alpha\ g_i^\mathcal{Q} \cdot g_i^\mathcal{U}\right)\right) - m_i \alpha\ g_i^\mathcal{Q} \cdot g_i^\mathcal{U}\right)$$

with $w_i$ a positive real number useful when match and non-match are unbalanced. In the following experiment we ignored this term since the task is way to unbalanced and adding a weighting term would break the loss function.

The $\beta$ term, in the tanh, is first set to 1 and then increased when a convergence criteria is obtain. This process repeats ten times, creating a sequence of of increasingly harder optimization which, if repeated infinitely, would converge to a discrete optimization.

$$\lim_{\beta \to \infty} \tanh(\beta x) = \text{sign}(x)$$

In all experiments, we use $\alpha = 0.2$ for the sigmoid to have enough signal in the range $[-32, 32]$.

### C.4 HAMMING RADIUS SEARCH

The Hamming Radius Search (HRS) is a naive approach to quickly find documents indexed with a binary code which have low Hamming distance r (the radius) to a particular binary code (the query). For insertion, we map each document's binary code the the document using a hash table. For searching, we compute all binary codes at distance than 0 (i.e. only the query) and use the hash table, then we do the same for distance up to r, the radius.

The number a binary codes to consider grow very quickly w.r.t to the radius. For example, at radius 2 with 64 bits codes, the number of codes to consider is 2081.

$$\binom{64}{0} + \binom{64}{1} + \binom{64}{2} = 2081$$

and for the radius 3 we there is 43745 codes to consider. Radius 3 would not make sense for a database of 10K elements as the last 41664 elements comes in no particular order. This is why we consider for the following experiments 2081 hash before halting for Hamming Radius Search.

# D SUPPLEMENTARY FIGURES

Table 2: The comparative table of SSWR's Halting Percentage

| Models | Unshared | | Shared | |
|---|---|---|---|---|
| | HRS | HMBS | HRS | HMBS |
| Fbeta | 22.51 % | 0.20 % | 8.64 % | 0.00 % |
| MIHash | 92.72 % | 92.19 % | 11.81 % | 0.10 % |
| HashNet | 100.00 % | 100.00 % | 16.14 % | 12.88 % |

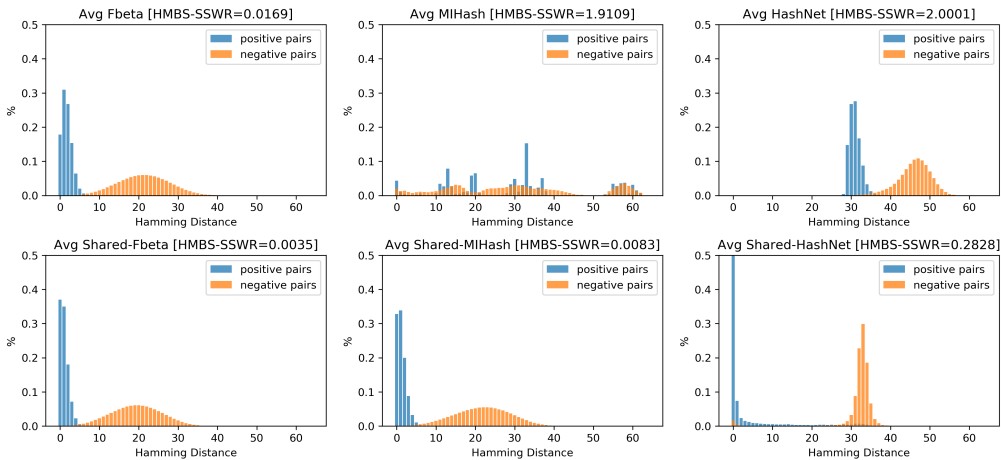

Figure 2: The average Hamming distance of the 25 models of each 6 HMBS models w.r.t. the positive (matching) pairs and negative (non-matching) pairs. Using the fixed 10K documents and 1K queries, creating 1K positive pairs and 9999K negative pairs for which we computed the Hamming distance.

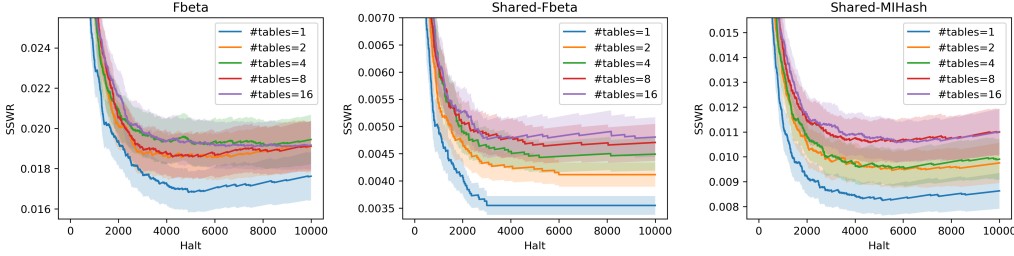

Figure 3: The average SSWR Curves W.r.t Halt number for Fbeta, Shared-Fbeta and SHared-MIHash. The colored area is ±0.01xSTD of the respective curve. The range of the y axis is changing throughout each graphs, this could be misleading when comparing the STD. As a reference, the average STD is 0.1259, 0.0374 and 0.0864 for Fbeta, Shared-Fbeta and Shared-MIHash respectively.

