# OpenReview forum: "Search Data Structure Learning"
_ICLR.cc/2021/Conference — Reject_

### Official Review · AnonReviewer1 · 2020-10-27
**Interesting line of research; lacks sufficient clarity and positioning with respect to relevant existing literature**

**Rating:** 3
**Confidence:** 4

**Review:**

This paper proposes a new objective to learn representations that allow for efficient and accurate search with the Multi-Bernoulli Search data structure. The motivation for such a scheme is strong and the preliminary empirical results demonstrate the utility of using the proposed representation learning objective and the data structure.


However, I am currently leaning towards a reject because of two main reasons: The first being the clarity of the presentation in the paper that makes it hard to identify (i) novel contributions, (ii) key algorithmic and technical details -- the main paper is supposed to be somewhat self-sufficient; with the current version, it was significantly hard for me to follow through the presentation even when repeatedly referring to the supplement. The second reason is the lack of positioning of the proposed scheme/objective/data structure to a long line of research on the use of machine learning (with novel metrics/objectives and data structures) for search, including the learning of space partitioning trees [A, B, C], locality sensitive hashes [E] and representations [D] -- it is quite possible that the proposed scheme is solving a different and/or more general problem but I believe it would be useful to connect this "Search Data Structure Learning" to the existing work on "Learning to Search".


Beyond the aforementioned high level comments, please find the following specific comments/questions that should be addressed:

- The set of relations $\mathcal{R}$ and the per-database $\mathcal{R}_D$ could use further motivation as to how they relate to search and/or relate to the task of learning search data structures.
- It is not clear why the distinction between "absolute" and "relative" is necessary in the context of the problem being targeted in this paper.
- The SSWR definition is unclear to me and can use more exposition. It is not clear that $\delta_t, t = 1, \ldots, T$ is a sequence of sets. Also, it is not explained why the oracle costs are only incurred for the final $\delta_T$ and not all intermediate ones (unless $\delta_{t-1} \subseteq \delta_t$) or to the complete $\cup_{i=1}^T \delta_t$.
- At the end of section 3, after presentation of SSWR, it is not clear why we are minimizing for a search sequence generator $\mathcal{G}$ that is aggregated over database-query pairs $(D, q)$ -- wouldn't we learn a data structure per database (as is done for data structures used for nearest-neighbor search)? Can this be clarified, and if it is deliberate, please motivate the need and advantage of learning across databases.
- The algorithm description is very hard to follow -- there are M back-end data structures and also the learned representation is used to generate M probable back-ends to insert in and T probable back-end to search from. It is not clear how the search involves $T$ passes over the $M$ back-ends. It would be good to clarify in the main paper that the "outcomes" are keys for each backend and all $T$ keys are tried in all $M$ back-ends.
- How are we leveraging multiple databases as shown in the previous loss function?
- The halting mechanism requires better presentation and clarification. It makes intuitive sense. But the presentation of the training algorithm (at least in the main paper) seems insufficient to provide enough context about the specifics of the halting mechanism.



[A] Li, Z., Ning, H., Cao, L., Zhang, T., Gong, Y., & Huang, T. S. (2011). Learning to search efficiently in high dimensions. In Advances in Neural Information Processing Systems (pp. 1710-1718).
[B] Cayton, L., & Dasgupta, S. (2008). A learning framework for nearest neighbor search. In Advances in Neural Information Processing Systems (pp. 233-240).
[C] Dong, Yihe, et al. "Learning Space Partitions for Nearest Neighbor Search." ICLR 2020.
[D] Sablayrolles, A., Douze, M., Schmid, C., & Jegou, H. (2018, September). Spreading vectors for similarity search. In International Conference on Learning Representations.
[E] Wang, J., Liu, W., Kumar, S., & Chang, S. F. (2015). Learning to hash for indexing big data -- A survey. Proceedings of the IEEE, 104(1), 34-57.

---

> ### Author Response · Authors · 2020-11-19
> **Re: Interesting line of research; lacks sufficient clarity and positioning with respect to relevant existing literature**
>
> Thanks a lot for the review. While reading it, we realized that we did not adequately explain and motivate SDSL's two fundamental aspects.
> * Dynamic databases
> * Relative relations
>
> First, in dynamic databases, we have to deal with insertion and deletion. The following extreme but tangible example of dynamic databases highly inspired us: a company wants to design a search engine trained once and ready for distribution to multiple clients without further training on each client's database. Hopefully, this showcases why we believe it is essential to have a framework that considers dynamic databases.
>
> Secondly, by relative relations, we mean the case where the relevance of a document w.r.t. a query depends on the other documents in the database. The most studied relative relations is probably KNN. It is relative since it is impossible to know if a document is in the k-nearest neighbour of a query without knowing the other documents. In contrast, radius search is an example of an absolute relation because it is possible to know if a document is relevant to a query only by looking at the query-document pair. However, we did not introduce relative relations only for KNN. Another rather exciting example is multiple supporting facts
> "A harder task is to answer questions where two supporting statements have to be chained to answer the question [...] where to answer the question "Where is the football?" one has to combine information from two sentences "John is in the playground" and "John picked up the football". "[1]
> but for a vast amount of facts.
>
> We propose to add, in the introduction, similar paragraphs as the ones above to motivate and clarify the two critical points of the SDSL framework.
>
> To put in contrast with the Learning to Search [A] framework. It is possible to update the framework to deal with dynamic databases by taking an expectation over the databases in the retrieval quality Q(t) and computational cost C(t). However, it is not clear how to deal with relative relations because the selection function T(q, x) is a "matching function" that does not exist for relative tasks. Generalizing the selection function by allowing it to consider the whole database (i.e. with T(q, X)) does not work because T(q, X) could use the ranking function s(x, q) on every document and nothing would penalize such exhaustive strategies since the computational cost is the number of candidates. Nevertheless, this is not the main issue. As with the framework proposed in [B], the computational cost does not consider the retrieval cost but only the size of the candidates set (divided by the number of documents in the database for [B]). Those frameworks fail to quantify the work needed to retrieve the candidates. For example, [A] relies on timing in the evaluation of their model. The SSWR is a unique quantity that quantifies both the cost of retrieval and the candidates' quality simultaneously.
>
> In the related work section, we propose to add a paragraph similar to the one above to position ourselves and clarify our framework.
>
> In the second part of this article, we propose a hashing based SDS that works only for absolute relations. While this is not tackling the most general case, we proposed the first instance (to the best of our knowledge) of an efficient hashing structure called Hashing Multi-Bernoulli Search (HMBS).
>
> "Specifically, the goal of learning to hash is to embed data into compact binary codes so that the hamming distance between two codes reflects their original similarity. In order to perform efficient hamming distance search using the embedded representation, an additional efficient algorithmic structure is still needed. (How to come up with such an efficient algorithm is an issue usually ignored by learning to hash algorithms.)" [A]
>
> We propose to add a concrete example with small values of n, M, and T in the appendix to solidify the description of the HMBS data structure. This is the kind of algorithm that is easier to explain with an example.
>
> Since we designed the proposed loss function for absolute relations, we only need to train with query-document pairs, i.e. we do not need the relevance to be relative to a database. This simplifies the minimization over query-database pairs to query-document pairs.
>
> Those modifications hopefully strengthen the article by making it more straightforward and better positioned within the current literature.
>
> [1] Weston, J., Bordes, A., Chopra, S., Rush, A. M., van Merriënboer, B., Joulin, A., & Mikolov, T. (2015). Towards ai-complete question answering: A set of prerequisite toy tasks. arXiv preprint arXiv:1502.05698.

---

> > ### Author Response · Authors · 2020-11-19
> > **Re: Interesting line of research; lacks sufficient clarity and positioning with respect to relevant existing literature (part 2)**
> >
> > To further clarify our framework:
> > 1) We propose to add: "This is to allow the general case where the relation is relative" after describing $R$ as a set of $R_D$.
> > 2) In this work, we provide an algorithm that is limited to the absolute case. However, the formalism that we introduced can describe a more general case that was never tackled before from the efficiency perspective.
> > 3) We propose to:
> > a) replace the first sentence of the paragraph preceding definition 3.5 with "It is not rare that an SDS can be slightly modified to produce a sequence of sets candidates.".
> > b) add "The reason why the SSWR uses the relevance oracle cost over the last sets of candidates only is because the generator did not found enough relevant documents before generating the last sets of candidates. Consequently, an exhaustive search with the oracle in the previous candidates sets was required before asking the generator to yield more candidates. The SSWR account for this exhaustive search with the |H| term. Finally, the sets of candidates are intended to be mutually exclusive because this will reduce the relevance oracle cost computed over the last sets of candidates and give a better SSWR. However, it is not mandatory." after the definition of the SSWR.
> > 4) We propose to add: "Where we minimize the expectation over all databases to ensure the quality of the generator even if the database changes. By letting the distribution over D to be deterministic, we fall into the case with a static database." after defining the objective.
> > 5) We propose to add: "It is possible to generate candidates (gen(q)) with MBS by yielding candidates each time a search in a back-end structure is made for the equivalent search(q)" in the paragraph explaining insertion and searching in MBS. We would also add generation with halting in the example given in the appendix.

---

### Official Review · AnonReviewer2 · 2020-10-28
**This paper proposes a general search framework called Search Data Structure Learning. A learning based method is designed to learn how to search in the database. In order to train the whole process, a novel loss function is designed and differentiable parameterized networks are adopted. The authors also design a novel metric to evaluate the performance. Experiments on NoisyMnist are conducted to demonstrate the effectiveness of the proposed method.**

**Rating:** 4
**Confidence:** 3

**Review:**

The paper is generally well written, and the authors target on an important and challenging problem of learning how to search. Search is a complex problem, and people designed various data structures to handle different kinds of search problems. The authors generalize these searching data structures by search data structure learning. The idea is novel and very interesting.

Then, the authors design SSWR metric evaluate the efficiency and quality of the search process. And the authors design ELBA(Efficient Learning Binary Access) by proposing a novel loss function and neural networks to create a discrete binary code(s) for both the queries and documents. And the authors conduct several experiments on NoisyMnist dataset to compare with several methods to show the advantage of the proposed framework.

MNIST dataset is a simple and small dataset, so more challenging datasets (such as SIFT-1M and GIST-1M) used in the search community are suggested to make the paper more convincing.

=======================

After reading all the review comments and rebuttals, I would like to change my score to 4.
The paper is interesting, but more detailed analysis and experiments are needed to make the work more clear and convincing.

---

### Official Review · AnonReviewer4 · 2020-10-29
**A new evaluation metric and a new loss for search via binary codes**

**Rating:** 4
**Confidence:** 4

**Review:**


This article presents a new evaluation metric that improves upon hamming distance of two. Meanwhile, a new F-beta loss is proposed with advantages over other learn to hash methods in this evaluation metric.



Strength:
A new formulation on both evaluation metric and a new loss function


Weakness:
1. Claim: In the introduction of the paper, different data structures such as graph-based Efanna, HNSW, and ONNG. However, in the following sections of the paper, the author does not discuss the possibility of SWR pr SSWR on these methods. Is SSWR only designed for binary codes based information retrieval?
2. Definition of cost: In Appendix A, the cost is estimated by how many calls in expectation
are needed to find those k elements if we sample from S uniformly without replacement. Is there any justification for why we assume the k elements by uniform sampling? In hashing based retrieval, the sampling is based on a distribution with PDE monotonic to the distance [1]. In this case, is the cost estimated correctly?
3. Comparison with other evaluation metrics: Is there any discussion of SSWR with evaluation metrics other than p@2. For instance, mAP, queries per second(QPS), and speedups over brutal force search. Is higher QPS or higher speedups over brutal force equivalent to higher SSWR?
4. Experiments: This paper only presents results in image search settings.  Is there more comparison of f-beta loss in different search settings? For instance, (1) ANN search with l2 or cosine similarity. (2) Maximum Inner Product Search (3) Cross-modal retrieval such as text-image search.

[1]Mutual Information Estimation using LSH Sampling https://www.semanticscholar.org/paper/Mutual-Information-Estimation-using-LSH-Sampling-Spring-Shrivastava/7241d1e839fdb69f7f0cc70220ad055e8900946c

---

> ### Author Response · Authors · 2020-11-19
> **Re: A new evaluation metric and a new loss for search via binary codes**
>
> 1) The SSWR (and SWR) is not designed only for binary codes and hash-based SDS. While it is possible to compare a graph-based SDS with another graph-based SDS, comparing a graph-based SDS with a hash-based SDS is tricky. In practice comparing time is possibly the ideal measure of cost. However, it is not adequate for theoretical analysis since the results depend on the implementation and the hardware used. Consequently, it is impossible to compare hash-based SDS with tree/graph-based SDS without making dubious assumptions (e.g. x hashes = y nodes lookups) that would be hard to justify and probably cause prejudice to one type of SDS.
> 2) The SSWR never evaluates the order the sets of candidates are generated using the relevance oracle cost because, indeed, the sets are not given randomly. The SSWR uses the relevance oracle cost on the generator's last set to quantify a random search cost in this set with the oracle.
> 3) We propose to add the following discussion of other metrics before the introduction of the SSWR.
> a) The mAP does not consider the work done to perform the ranking. An SDS could compare the query to every document in the database and have a good mAP. In SDSL, we want to monitor the quality as well as the efficiency of retrieval.
> b) In our opinion, the Recall Query per second (RQPS) is not suitable for theoretical analysis since it depends on the implementation and hardware used. However, it would be easy to generalize the RQPS by changing what quantifies the work (something other than the number of seconds). However, the most crucial contribution of SSWR w.r.t. the RQPS is that it allows the SDS to generate an arbitrary number of candidates. This is because the RQPS has a parameter (k) that limits the number of candidates it generates. This parameter prevents a model from generating all documents in the database as its candidates, which would give 100% recall without any computation. Ultimately, the parameter k is a fix to the flaw that the RQPS does not consider the precision. Producing the database is not viable to have a good SSWR, and the SSWR achieves this without limiting the number of candidates that can be generated.
> c) The SSWR is the speedup over brutal search when we answer the following questions. What happens when the searches are not exact? How to deal with candidates of varying sizes? How to take into account searches that do not find enough relevant documents?
> d) The SSWR is probably correlated with the RQPS, but it is possible to (adversarially) design models where they are not. For example, by randomly generating the first k candidates.
> 4) We were not particularly interested in ANN as we aim to show that it is possible to work in a generalized framework. For cross-modal retrieval, while the SDSL framework supports it. Hashing is inadequate, in general. This is because hashing implies some generalized form of transitivity in the relation, i.e. q1 ~ d1, q2 ~ d1, q2 ~ d2 ⇒ q1 ~ d2, which does not hold for most cross-modal retrieval tasks. Cross-modal retrieval is part of our field of interest but would probably require something else than a hashing based SDS.

---

### Official Review · AnonReviewer3 · 2020-10-29
**The paper needs to justify the new metric.**

**Rating:** 4
**Confidence:** 4

**Review:**

In this paper, the authors proposed Search Data Structure Learning (SDSL), which they claim to be a generalization of the standard Search Data Structure. They also present a new metric called Sequential Search Work Ratio (SSWR) to evaluate the quality and efficiency of the search. They introduced a new loss called F-beta Loss, showing their algorithm is better than two previous results, MIHash (Cakir et al. 2017) and HashNet (Cao et al. 2017).

I appreciate the key message the paper is trying to convey: we need the formal definition or mutual agreement on the problem as well as the correct evaluation metrics to push forward a research area. However, I have several major concerns about the contribution of this paper.

1. I do not see a formal definition of SDSL. Definition 3.1 is just a definition of matching and non-matching relations.
2. What is new in the metric defined in Definition 3.4? The denominator is constant for all search methods. C(.,.,.) is the cost of re-ranking, and w0 is the cost of searching (filtering the candidate).
3. There is no theoretical or empirical comparison/evaluation of the proposed metric. The calculations in Appendix A are very standard calculations. It is unclear about the innovation of this metric from a theory perspective. In experiments, the authors directly use SSWR, and there is no justification on why this is the right metric.

Before proposing the new loss, etc., I suggest the authors could go back and justify the metric's effectiveness. Otherwise, it is hard to conclude if this paper has made any progress.

=========================================

Thanks for the rebuttal and revision. My first concern has been addressed. However, I still found the proposal lack empirical or theoretical proof, so I am not convinced the contribution is principle enough. I decide to keep my original score.

---

> ### Author Response · Authors · 2020-11-19
> **Re: The paper needs to justify the new metric.**
>
> 1) Definition 3.1 does not define the SDSL framework. It formalizes the dichotomy between relative and absolute relations. The definition of SDSL is the minimization at the end of section 3. To clarify, we propose to add a formal definition above the last paragraph of section 3, which will be modified to support the definition.
> 2)
> a) The denominator is constant since it is the cost of using the oracle without the SDS. For comparing models, it is possible only to use the numerator. However, it is tough to interpret the SDS quality only by looking at the numerator. The SSWR is a ratio with a straightforward interpretation. E.g. a SSWR of 0.5 indicates that the SDS is twice as performant as the oracle alone. Furthermore, an SSWR below 1 indicates that the SDS is useful.
> b) The relevance oracle cost is not the cost of re-ranking. It considers how many relevant documents are present in a set and how many are required and compute the expected number of calls to the oracle needed to find enough relevant documents by doing a random search. We can interpret the |H| term in the SSWR as a re-ranking cost of the previous candidates' sets.
> 3) The SSWR compares the scenario where we use an oracle and an SDS with the scenario where we use an oracle only in a way that considers both the retrieval's quality and cost simultaneously.

---

### Decision · Program_Chairs · 2021-01-07
**Final Decision**

**Decision:**

Reject

**Comment:**

This paper addresses an interesting problem and all reviewers agree.  Most reviewers found the paper difficult to understand and it was hard to see the novel contributions.    The paper will need a significant revision before publication.